# Electrically Inspired Flexible Electrochemical Film Power Supply for Long-Term Epidermal Sensors

**DOI:** 10.3390/mi14030650

**Published:** 2023-03-13

**Authors:** Hao Zheng, Xingguo Zhang, Chengcheng Li, Wangwang Zhu, Dachao Li, Zhihua Pu

**Affiliations:** State Key Laboratory of Precision Measuring Technology and Instruments, Tianjin University, Tianjin 300072, China; zhenghao_0402@tju.edu.cn (H.Z.);

**Keywords:** film power, electrically inspired, electrochemical, epidermal sensors

## Abstract

This paper, for the first time, reports an electrically inspired flexible electrochemical film power supply for long-term epidermal sensors. This device can periodically provide electrical power for several hours after a short-time electrical activation. The electrical activation makes acetylcholine, which is infused into the subcutaneous tissue by iontophoresis. The interstitial fluid (ISF) with glucose molecules is then permeated autonomously for several hours. At this period, the device can provide electrical power. The electrical power is generated from the catalyzing reaction between the glucose oxidase immobilized on the anode and the permeated glucose molecules. After the ISF permeation stops, we give a short-time electrical activation to provide electrical power for several hours again. The power supply is flexible, which makes it adaptively conform to skin. The episodic short-time electrical activation can be enabled by an integrated small film lithium-ion battery. This method extends the service life of a lithium-ion battery 10-fold and suggests the application of small lithium-ion batteries for long-term epidermal sensors.

## 1. Introduction

Recently, with the development of mechanics, materials and manufacturing processes, epidermal electronic devices have been actively studied [1]. The flexible electronic devices can be attached to human skin, just like a second skin of the human body. Named electronic skin, it is a fully integrated electronic system composed of multifunctional electronic components [2]. Flexible sensors in electronic skin make it possible to continuously, and even non-invasively, function as citizen medicine and provide home healthcare and disease prevention [3]. There has been much research on, and even products that include, epidermal sensors for monitoring information such as body and skin temperature [4], body movement [5], electrophysiological activity [6]; conducting molecular analyses of sweat [7] and interstitial fluid [8]; wirelessly communicating the collected information; and then connecting the human body to the internet. This will be a necessary way of incorporating the human body into the internet [9]. However, the power supply of the epidermal electronic devices is still a challenge.

Photovoltaics [10,11] and triboelectric and piezoelectric nanogenerators [12,13] have been explored to provide energy for epidermal electronics. Lam et al. developed efficient textile-based flexible perovskite solar cells by employing a low-temperature electrodeposited tin oxide electron-transporting layer coupling, which could be an efficient power supply system for wearable electronic devices [14]. Despite rapid developments in recent years, photovoltaic energy for flexible wearable devices still suffers from insufficient energy density, difficulties in large-scale device fabrication [11], and more importantly, from being limited by environmental illumination conditions [15]. Wang et al. reported the first experimental study for the piezoelectric performance of monolayer MoS_2_ and showed a voltage output of 15 mV, current output of 20 pA and power density of 2 mW·m^−2^ under bending of 0.53% [16]. Nanogenerators have grown rapidly in recent years, but must rely on friction or moving components [17]. Photovoltaics and nanogenerators are promising, but constrained by illumination conditions and sustained activation; thus, they are inappropriate for long-term epidermal sensors. Flexible lithium-ion batteries [18] have been developed for wearable electronics in recent years. However, in order to service them for long-term sensors, a large area or frequent replacement is required.

Biofuel cells (BFC) are a power source that use a biocatalyst as an electrode catalyst. BFC can convert energy in organic matter, such as sugars and alcohols, into electricity, and have the advantage of being stable, safe and high-energy-density energy carriers. BFCs are considered a power source for wearable health monitoring devices [19,20]. Shitanda et al. proposed a BFC that can be incorporated into diapers to be powered by glucose in urine [21]. Sweat is extremely low in glucose, but the lactic acid in it is also a great source of fuel [19]. Wang et al. developed a tattoo-type BFC with low wearing sensation that can generate approximately 40 μW/cm^−2^ of electricity using the lactate in sweat [22]. The fuel supply of these BFCs relies on the passive exudation of body fluids, which often limits the use conditions of energy and cannot be carried out in some special environments. Zhang et al. proposed an epidermal electrochemical energy source with a replaceable glucose membrane [23]; however, replacing the glucose membrane is tedious and heavily reliant on professionals.

In this paper, an electrically inspired flexible electrochemical film power supply is proposed for long-term epidermal sensors. The electrical activation infuses acetylcholine into the subcutaneous tissue by iontophoresis. The interstitial fluid (ISF) with glucose molecules is then permeated autonomously. Glucose in the exuded ISF provides energy for the continuous supply. This power supply can continue to provide power for hours after short-time electrical activation. Moreover, this power supply is flexible and fits well with the skin.

## 2. Methods and Materials

**Design and mechanism.** As shown in Figure 1a, an electrically inspired flexible electrochemical film power supply is designed for long-term epidermal sensors. This device can periodically provide electrical power for several hours after a short-time electric activation. This power supply consists of two electrodes, for which structures are demonstrated in Figure 1b. The anode of the energy source consists of a silver/silver chloride electrode layer and a chitosan layer mixed with GOx, platinum nanoparticles (PtNPs) and acetylcholine. The cathode of the energy source consists of a silver/silver chloride electrode layer and a chitosan layer mixed with PtNPs and laccase. Glucose is catalyzed by GOx to generate electrons [8]. PtNPs can improve the electron transfer efficiency, thereby increasing the current density of film power supply [24]. After acetylcholine enters the skin by iontophoresis, it causes subcutaneous vasodilation [25] and changes skin permeability, allowing ISF to leak out. For the application, the acetylcholine on the anode is first infused into the subcutaneous tissue by iontophoresis via short-time electrical activation to make ISF permeate autonomously for the next several hours. Then, as illustrated in Figure 1c, this power supply can independently generate power from glucose molecules, which are completely oxidized to carbon dioxide and water by GOx and PtNPs on the anode; meanwhile oxygen is reduced to water by laccase and PtNPs on the cathode. Polyimide (PI) film was selected as the substrate because of its good insulation properties, good flexibility and ease of production, as demonstrated in our previous work [8]. Electrochemical data of the electrodes were collected using an electrochemical workstation (CHI852D, Shanghai Chenhua Instrument Co., Ltd., Shanghai, China). The cells were electrically excited using a potentiostat (ZF-9, Shanghai Fangzheng Electronics Co., Ltd., Shanghai, China).

After acetylcholine penetrates the skin, the glucose in the exuded ISF undergoes an oxidation reaction under the catalysis of GOx in the anode and loses electrons. The electrons are transferred from the anode to the cathode. Laccase in the cathode catalyzes the decomposition of oxygen using the obtained electrons. The current flows from the cathode to the anode. Finally, glucose and oxygen react into carbon dioxide and water, thereby realizing the conversion of chemical energy into electrical energy. The theoretical reaction formula is as follows:Anode: C_6_H_12_O_6_ + 24OH^−^ → 6CO_2_ + 18H_2_O + 24e^−^(1)
Cathode: 6O_2_ + 12H_2_O + 24e^−^ → 24OH^−^(2)
Overall: C_6_H_12_O_6_ + 6O_2_ → 6CO_2_ + 6H_2_O(3)
ΔG = −2.870 × 10^−6^ J/mol, U = 1.24 V

If glucose is fully oxidized, 1 mol of glucose can release 2.87 MJ, that is, 15.94 kJ per gram of glucose. Under complete oxidation, each glucose molecule reacts with 24 electrons, making glucose a good choice for BFC fuel [26].

**Fabrication and modification.** Figure 2 shows a schematic diagram of the electrochemical film power supply fabricating processes, including substrate preparation, electrode printing and electrode modification. The polydimethylsiloxane (PDMS) (Sylgard 184, Dow Corning Co., Ltd., Midland, TX, USA) was first dropped on a glass substrate. The flexible substrate was then fabricated by spinning PI (Sigma Aldrich Inc., Saint Louis, MI, USA) on the PDMS, as shown in Figure 2a. A silver/silver chloride paste (C2130809D5, Gwent Co., Ltd., Torfaen, UK) was then screen-printed onto the PI substrate and heated at 80 °C for 15 min to make 2 conductive electrodes (Figure 2b). After that, a mixture of 20 wt % GOx (Tianjin Jiangtian Huagong Co., Ltd., Tianjin, China) aqueous solution, acetylcholine solution, 1.21 mM PtNPs (Changchun Institute of Applied Chemistry Chinese Academy of Sciences, Changchun, China) aqueous solution and 0.5 wt % chitosan aqueous solution (1:1:1:10 by weight) was dropped on the anode. A mixture of 20 wt % laccase (Sigma Aldrich Inc., Saint Louis, USA, from Trametes Versicolor) aqueous solution, 1.21 mM PtNPs and 0.5 wt % chitosan aqueous solution was dropped on the cathode (1:1:10 by weight) (Figure 2c). Finally, Nafion solution was dropped on the patches after the two mixtures were solidified at room temperature (Figure 2d). The flexible electrochemical film power supply could be obtained by peeling from PDMS.

## 3. Result and Discussion

Scanning electron microscope (SEM) images of the anode and cathode of the fabricated BFC are shown in Figure 3a,b. SEM images show that the BFC has a uniform structure and a smooth surface. Figure 3c,d are X-ray photoelectron spectroscopy (XPS) images of the anode and cathode of the fabricated BFC. XPS results show that PtNPs, glucose, acetylcholine, GOx and laccase have 3D nanostructures.

The fabricated BFC were tested in vitro, and all experiments were conducted at room temperature in the laboratory. The concentration of glucose solution used in in vitro experiments was 180 mg/dL, and the solvent used was 0.1 M Phosphate buffer solution (PBS, Tianjin Jiangtian Huagong Co., Ltd., Tianjin, China). All data were measured with an electrochemical workstation. The open circuit voltage of the BFC was tested, as shown in Figure 3e, and the results showed that the average open circuit voltage of the BFC was 151 mV, the maximum voltage was 155 mV and the minimum voltage was 141 mV at a time of 400 s. The voltage was relatively stable. Figure 3f shows the current output of the BFC, with an average current of 12.7 μA, measured using chronoamperometry over a period of 400 s. The two power sources were connected in parallel, and the chronoamperometry was used for detection. Figure 3g shows the output current of the power supply when two BFCs are connected in parallel. The average current value reaches 20.6 μA, 1.6 times that of a single power supply. Figure 3h shows the current output of the BFC for a long time, validating the requirement that the proposed BFC can achieve long-time function. The in vitro experiments proved the feasibility of the proposed BFC. When the BFCs are connected in parallel, the power supply current increases significantly. Therefore, the BFC can be made into an array form in the future to increase the power in order to meet the needs of flexible electronics.

To verify the effect of the proposed BFC, in vivo experiments were performed using rats. The rat was anesthetized and placed on the operating table, and its abdominal hair was shaved. The fabricated BFC was attached to the rat’s abdomen and fixed with tape, as shown in Figure 4a. The BFC was briefly excited by a constant voltage source, acetylcholine infiltrated into the subcutaneous tissue under the action of the electric field force, and the glucose in the body began to exude. Initially, the voltage of the fabricated BFC was measured using the open circuit voltage method without electrical activation. The experimental results are shown in Figure 4b, and the open-circuit voltage was maintained at ~30 mV. The excitation voltage was adjusted to 150 mV and the excitation was continued for 3 min. After waiting for 5 min, the output voltage of the BFC was detected by the electrochemical workstation. Figure 4c shows the voltage output of the BFC after an excitation voltage of 150 mV. Its output voltage gradually increased from 75 mV to 100 mV, and achieved a stable voltage output for 5000 s. The proposed BFC can be powered spontaneously by the ISF through short-term electrical activation: reduce the voltage of excitation voltage to 40 mV and excite continuously for 3 min. After waiting for five minutes, the open-circuit voltage of the BFC was recorded, as shown in Figure 4d. The voltage stays at ~90 mV for almost 1000 s and then starts to decrease. We believe this is because the osmotic effect of acetylcholine has been exhausted, and the complex barrier properties of the skin are gradually restored. Excitation was performed again with a voltage of 40 mV for 3 min. We waited five minutes and measured the battery’s open circuit voltage again. Its voltage output was slightly smaller than that of the first time, but still close to 90 mV, and the result is shown in Figure 4e. Connecting an op-amp to the BFC, a light-emitting diode (LED) was lit up based on the proposed energy source, as shown in Figure 4f. In the later stage, lithium batteries can be used as excitation sources instead. The episodic short-time electrical activation can be enabled by an integrated small film lithium-ion battery. This method extends the service life of a lithium-ion battery 10-fold and suggests the application of lithium-ion batteries for long-term epidermal sensors.

## 4. Summary

In summary, we demonstrate an electrically inspired flexible electrochemical film power supply for long-term epidermal sensors. The experimental results show that the glucose permeated from the subcutaneous tissue can continue to provide energy for several hours. This flexible film power supply can be securely attached to the skin. In the future, the episodic short-time electric activation can be enabled by an integrated small film lithium-ion battery. The method provides the idea for the use of small lithium-ion batteries in long-time use epidermal sensors.

## Figures and Tables

**Figure 1 micromachines-14-00650-f001:**
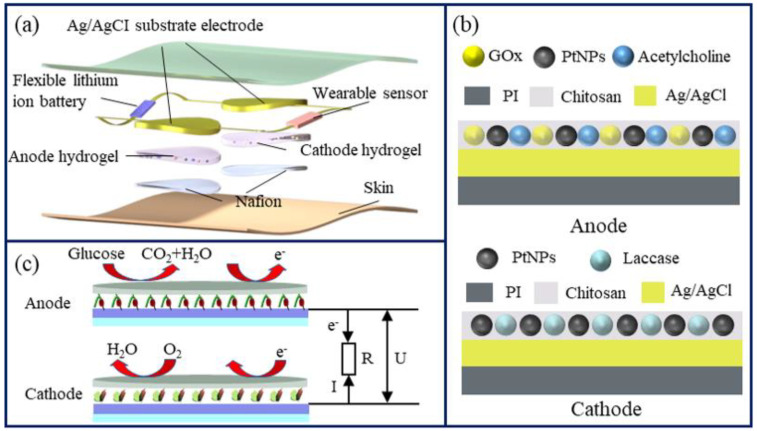
(**a**) Schematic of the electrically inspired flexible electrochemical film power. (**b**) Compositions of the anode (up) and cathode (down). (**c**) Mechanism of the flexible electrochemical energy source.

**Figure 2 micromachines-14-00650-f002:**
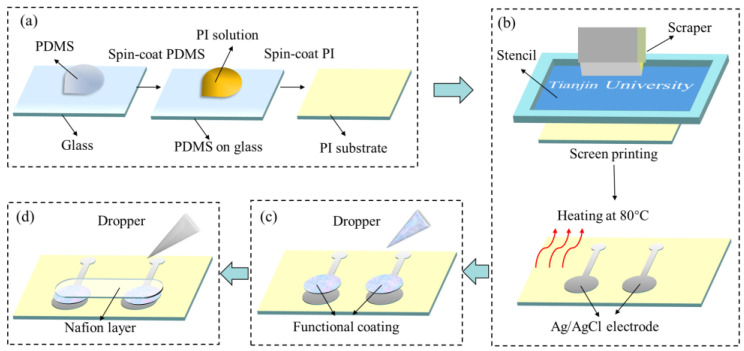
(**a**) Schematic diagram of PI substrate fabrication. (**b**) Schematic diagram of electrode fabrication. (**c**) Schematic diagram of functional layer modification. (**d**) Schematic diagram of Nafion layer modification.

**Figure 3 micromachines-14-00650-f003:**
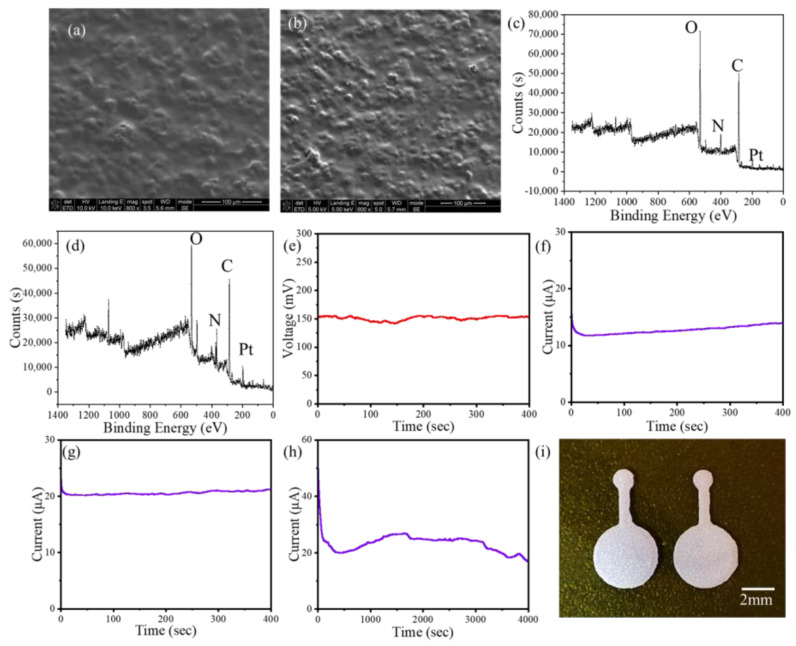
(**a**) SEM image of the anode. (**b**) SEM image of the cathode. (**c**) XPS image of the anode. (**d**) XPS image of the cathode. (**e**) Voltage output of the single BFC in 400 s. (**f**) Current output of the single BFC in 400 s. (**g**) Current output of the two BFCs paralleled in 400 s. (**h**) Current output of the two BFCs paralleled in 4000 s. (**i**) Photo of the electrode.

**Figure 4 micromachines-14-00650-f004:**
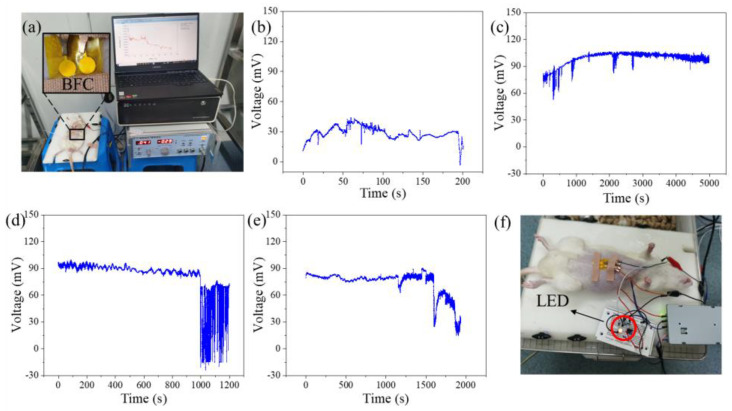
(**a**) Photo of the animal experiment. (**b**) Voltage output before electrical activation. (**c**) Voltage output after electrical activation of 150 mV. (**d**) Voltage output after first 40 mV electrical activation. (**e**) Voltage output after second 40 mV electrical activation. (**f**) Photo of an LED, lit up based on the BFC.

## Data Availability

Not applicable.

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
