# Peer review of "Electrically Inspired Flexible Electrochemical Film Power Supply for Long-Term Epidermal Sensors"

_micromachines, 2023, doi:10.3390/mi14030650_

Round 1

Reviewer 1 Report

Reviewer’s comments:

1.The typing text in Figure 1(a) and the text in the axis of both on Figure 3 and Figure 4 should be enlarged for the readability.

2.Current direction and potential polarity of a small lithium-ion battery for long-term epidermal sensors should be marked in Fig.1 to understand the operation of sensor.

3.The issue of the flexibility should be briefly described in the text on using the glass substrate with PDMA and PI films to match on the electric-inspired flexible electrochemical film power supply.

Author Response

Response to the reviewers' report:

Response to reviewer #1:

Dear reviewer #1:

Many thanks for your good questions and suggestions. The manuscript has been revised based on your good questions and suggestions (red font in the revised manuscript). The red words show the positions for each revision. Thanks for your contribution to the improvement of the manuscript.

COMMENT 1:

The typing text in Figure 1(a) and the text in the axis of both on Figure 3 and Figure 4 should be enlarged for the readability.

Our response: Thank you very much for your suggestion. It is a very good suggestion. We adjusted the typing text in Figure 1(a) and the text in the axis of both on Figure 3 and Figure 4.

We put these Figures here for your convenience.

"…

Figure 1. (a) Schematic of the electric-inspired flexible electrochemical film power. (b) Compositions of the anode (up) and cathode (down). (c) Mechanism of the flexible electrochemical energy source.

Figure 3. (a) SEM image of the anode. (b) SEM image of the cathode. (c) XPS image of the anode. (d) XPS image of the cathode. (e) Voltage output of the single BFC in 400 s. (f) Current output of the single BFC in 400 s. (g) Current output of the two BFCs paralleled in 400 s. (h) Current output of the two BFCs paralleled in 4000 s. (i) Photo of the electrode.

Figure 4. (a) Photo of the animal experiment. (b) Voltage output before electrical activation. (c) Voltage output after electrical activation of 150 mV. (d) Voltage output after first 40 mV electrical activation. (e) Voltage output after second 40 mV electrical activation. (f) Photo of a LED lighted up based on the BFC.

…"

COMMENT 2:

Current direction and potential polarity of a small lithium-ion battery for long-term epidermal sensors should be marked in Fig.1 to understand the operation of sensor.

Our response: Thank you very much for your suggestion. It is a very good suggestion. We marked the direction of the current flow and the direction of electron transfer in Figure 1(c), as well as the reactions occurring at the anode and cathode for the convenience of readers. We also added descriptions in the article, as shown in lines 97-99 of page 3.

We also put it here for your convenience.

"…Electrons are transferred from the anode to the cathode. Laccase in the cathode catalyzes the decomposition of oxygen using the obtained electrons. Current flows from the cathode to the anode…"

COMMENT 3:

The issue of the flexibility should be briefly described in the text on using the glass substrate with PDMA and PI films to match on the electric-inspired flexible electrochemical film power supply.

Our response: Thank you very much for your suggestion. It is a very good suggestion. We supplemented the description of the flexibility in the article, as shown in lines 68-69 of page 2, lines 87-89 of page 2, lines 112-115 of page2 and lines 124-125 of page 3.

We also put these Figures here for your convenience.

"…Moreover, this power supply is flexible and fits well with the skin.

…Polyimide (PI) film was selected as the substrate because of its good insulation properties, good flexibility and ease of production, as demonstrated in our previous work.8

The polydimethylsiloxane (PDMS) (Sylgard 184, Dow Corning Co., USA) was firstly dropped on a glass substrate, then the flexible substrate is fabricated by spinning PI (Sig-ma Aldrich Inc., USA) on the PDMS, as shown in Figure 2a.

The flexible electrochemical film power supply could be obtained by peeling from PDMS."

Reviewer 2 Report

In this manuscript, Zheng et al. firstly reports an electric-inspired flexible electrochemical film power supply for long-term epidermal sensors. The electrical power can be provided periodically for several hours after a short-time electric inspiring which enabled by a small film lithium-ion battery. Such method provides a way of a small lithium-ion battery for application in long-term epidermal sensors. This paper is interesting and has clear logic. It can be considered for the publication in this journal after revising the following minor issues:

1. Page 3 line 98-101: the sentence “After acetylcholine penetrates the skin, the glucose in the ISF exuded undergoes oxidation reaction under the catalysis of GOx in the anode and loses electrons.” is repeated.

2. Page 3, Figure 1b: As authors described “The anode of …chitosan layer mixed with GOx, platinum nanoparticles (PtNPs), and acetylcholine.” However, in Figure 1b, the fourth component “laccase” is also in the chitosan layer of the anode.

Moreover, Could you please give a brief description of the function of each component in Figure 1b?

3. Page 3, Figure 1c: As authors described “this power supply can independently generate power from glucose molecules which are completely oxidized to carbon dioxide and water by GOx and PtNPs on the anode, ...” However, Figure 1c does not match this description.

4. Abstract, the last paragraph of Introduction, and Summary have basically the same expression. Please polish up these sentences.

Author Response

Response to reviewer #2:

Dear reviewer #2:

Many thanks for your good questions and suggestions. The manuscript has been revised based on your good questions and suggestions (red font in the revised manuscript). The red words show the positions for each revision. Thanks for your contribution to the improvement of the manuscript.

General comment:

In this manuscript, Zheng et al. firstly reports an electric-inspired flexible electrochemical film power supply for long-term epidermal sensors. The electrical power can be provided periodically for several hours after a short-time electric inspiring which enabled by a small film lithium-ion battery. Such method provides a way of a small lithium-ion battery for application in long-term epidermal sensors. This paper is interesting and has clear logic. It can be considered for the publication in this journal after revising the following minor issues:

COMMENT 1:

Page 3 line 98-101: the sentence “After acetylcholine penetrates the skin, the glucose in the ISF exuded undergoes oxidation reaction under the catalysis of GOx in the anode and loses electrons.” is repeated.

Our response: Thank you very much for your suggestion. We removed this repeated sentence and reviewed the full article.

COMMENT 2:

Page 3, Figure 1b: As authors described “The anode of …chitosan layer mixed with GOx, platinum nanoparticles (PtNPs), and acetylcholine.” However, in Figure 1b, the fourth component “laccase” is also in the chitosan layer of the anode.

Moreover, Could you please give a brief description of the function of each component in Figure 1b?

Our response: Thank you very much for your suggestion. It is a very good suggestion. We corrected this mistake on Figure 1(b), and we added the description of the function of each component in Figure 1(b), as shown in lines 78-81 of page 2.

We also put the Figure here for your convenience.

Figure 1. (a) Schematic of the electric-inspired flexible electrochemical film power. (b) Compositions of the anode (up) and cathode (down). (c) Mechanism of the flexible electrochemical energy source.

COMMENT 3:

Page 3, Figure 1c: As authors described “this power supply can independently generate power from glucose molecules which are completely oxidized to carbon dioxide and water by GOx and PtNPs on the anode, ...” However, Figure 1c does not match this description.

Our response: Thank you very much for your suggestion. It is a very good suggestion. We corrected this mistake on Figure 1(c), as shown in the response to COMMENT 2.

COMMENT 4:

Abstract, the last paragraph of Introduction, and Summary have basically the same expression. Please polish up these sentences.

Our response: Thank you very much for your suggestion. It is a very good suggestion. We adjusted the structure of Abstract, the last paragraph of Introduction, and Summary, especially the part of the last paragraph of Introduction, and Summary. We polished these sentences and removed redundant sentences, as shown in lines 61-69 of page 2 and lines 191-197 of page 6.

We also put it here for your convenience.

"…In this paper, an electric-inspired flexible electrochemical film power supply is proposed for long-term epidermal sensors. The electric inspiring makes acetylcholine infused into the subcutaneous tissue by iontophoresis. Then the interstitial fluid (ISF) with glucose molecules will be permeated autonomously. Glucose in the ISF exuded provides energy for the continuous supply. This power supply can continue to provide power for hours after short-time electric inspiring. Moreover, this power supply is flexible and fits well with the skin…

…In summary, we demonstrate an electric-inspired flexible electrochemical film power supply for long-term epidermal sensors. The experimental results show that the glucose permeated from the subcutaneous tissue can continue to provide energy for several hours. This flexible film power supply can be well attached to the skin. In the future, the episodic short-time electric inspiring can be enabled by an integrated small film lithium-ion battery. The method provides an idea to use a small lithium-ion battery for long-time use epidermal sensors…"

Reviewer 3 Report

Manuscript ID: micromachines-2255683

Title: Electric-inspired flexible electrochemical film power supply for long-term epidermal sensors

This manuscript presents the preparation of flexible electrochemical films for the anode and cathode of biofuel cells (BFCs) as an energy source.

The manuscript is relevant and interesting. New systems are needed to deliver epidermal electronic energy. A biofuel cell (BFC) is an energy source that uses a biocatalyst as an electrode catalyst, converting energy from organic matter such as glucose into electricity.

The BFC was fabricated in this work and used alone and in an array of two connected in parallel as a power source for wearable health monitoring devices. Several physical methods were used to characterize the electrochemical films (SEM, XPS, chronoamperometry). To verify the effect of the proposed BFC, in vitro and in vivo experiments were performed.

The manuscript is well written, the experiments are well designed, and the results are well presented, but in a pile. I would recommend a more rigorous systematization.

Figures and tables support the authors' comments.

Please explain the role of acetylcholine.

Results and discussion

Pag. 3.

Line 99 - 101: Please delete the repeated phrase.

Author Response

Response to reviewer #3:

Dear reviewer #3:

Many thanks for your good questions and suggestions. The manuscript has been revised based on your good questions and suggestions (red font in the revised manuscript). The red words show the positions for each revision. Thanks for your contribution to the improvement of the manuscript.

General comment:

This manuscript presents the preparation of flexible electrochemical films for the anode and cathode of biofuel cells (BFCs) as an energy source.

The manuscript is relevant and interesting. New systems are needed to deliver epidermal electronic energy. A biofuel cell (BFC) is an energy source that uses a biocatalyst as an electrode catalyst, converting energy from organic matter such as glucose into electricity. 

The BFC was fabricated in this work and used alone and in an array of two connected in parallel as a power source for wearable health monitoring devices. Several physical methods were used to characterize the electrochemical films (SEM, XPS, chronoamperometry). To verify the effect of the proposed BFC, in vitro and in vivo experiments were performed.

COMMENT 1:

The manuscript is well written, the experiments are well designed, and the results are well presented, but in a pile. I would recommend a more rigorous systematization.

Figures and tables support the authors' comments.

Our response: Thank you very much for your suggestion. It is a very good suggestion. We carefully checked the descriptions of the figures and texts, and adjusted the descriptions to make the manuscript more systematic and readable.

COMMENT 2:

Please explain the role of acetylcholine.

Our response: Thank you very much for your suggestion. It is a very good suggestion. Acetylcholine is a neurotransmitter that dilates blood vessels under the skin, which alters skin permeability so that ISF can leak out. This kind of material has been used to help expel body liquids by several works as shown in references [1] and [2]. We added the role of acetylcholine and reference in the article to make the article more readable, as shown in lines 80-81 of page 2.

We also put it here for your convenience.

"…After acetylcholine enters the skin by iontophoresis, it causes subcutaneous vasodilation25 and changes skin permeability so that ISF can leak out…"

COMMENT 3:

Results and discussion

Pag. 3.

Line 99 - 101: Please delete the repeated phrase

Our response: Thank you very much for your suggestion. We removed this repeated sentence and reviewed the full article.

References:

  1. Gardner-Medwin, J. M.; Taylor, J. Y.;  Macdonald, I. A.; Powell, R. J., An investigation into variability in microvascular skin blood flow and the responses to transdermal delivery of acetylcholine at different sites in the forearm and hand. Br. J. Clin. Pharmacol. 1997, 43 (4), 391-397.
  2. Lee, J.-B.; Bae, J.-S.;  Matsumoto, T.;  Yang, H.-M.; Min, Y.-K., Tropical Malaysians and temperate Koreans exhibit significant differences in sweating sensitivity in response to iontophoretically administered acetylcholine. Int. J. Biometeorol. 2009, 53 (2), 149-157.
